# Parameter Optimization for Modulation-Enhanced External Cavity Resonant Frequency in Fiber Fault Detection

**Xiuzhu Li [1,2], Min Zhang [1,2], Haoran Guo [1,2], Zixiong Shi [1,2], Yuanyuan Guo [1,2], Tong Zhao [1,2],\*** and **Anbang Wang [3,4]**

[1] Key Laboratory of Advanced Transducers and Intelligent Control System, Taiyuan University of Technology, Ministry of Education and Shanxi Province, Taiyuan 030024, China; lixiuzhu1093@link.tyut.edu.cn (X.L.); zhangmin1157@link.tyut.edu.cn (M.Z.); guohaoran1488@link.tyut.edu.cn (H.G.); shizixiong0936@link.tyut.edu.cn (Z.S.); guoyuanyuan@tyut.edu.cn (Y.G.)

[2] College of Electronic Information and Optical Engineering, Taiyuan University of Technology, Taiyuan 030024, China

[3] Guangdong Provincial Key Laboratory of Photonics Information Technology, Guangzhou 510006, China; abwang@gdut.edu.cn

[4] School of Information Engineering, Guangdong University of Technology, Guangzhou 510006, China

\* Correspondence: zhaotong.tyut@outlook.com

**Abstract:** Fiber fault detection is crucial for maintaining the quality of optical communication, especially in well-established optical access networks with extended distances and a growing number of subscribers. However, the increasing insertion loss in fiber links presents challenges for traditional fault-detection methods in capturing fault echoes. To overcome these limitations, we propose a modulation-enhanced external-cavity-resonant-frequency method that utilizes a laser for fault echo reception, providing improved sensitivity compared to traditional photodetector-based methods. Our previous work focused on analyzing key parameters, such as sensitivity and spatial resolution, but did not consider practical aspects of selecting optimal modulation parameters. In this study, we develop a model based on Lang–Kobayashi rate equations for current-modulated optical feedback lasers and validate it through experimental investigations. Our findings reveal that optimal detection performance is achieved with a modulation depth of 0.048, a frequency sweeping range of 0.6 times the laser relaxation oscillation frequency, and a frequency sweeping step of 0.1 times the external cavity resonant frequency.

**Keywords:** fiber fault detection; high-sensitivity detection; optical feedback; modulation resonance effect; external cavity signature

## 1. Introduction

The development of optical access networks has accelerated the emergence of the "Internet of Things" era [1–5]. Consequently, ensuring the maintenance of optical fiber links has become increasingly crucial to prevent communication interruptions and associated losses [6]. An optical time-domain reflectometer (OTDR) [7] is commonly employed for detecting faults in fiber links. However, as transmission distance and the number of subscribers continue to increase, the inherent insertion loss in the transmission link also rises. For instance, in a single-mode fiber with a transmission distance of 100 km, the one-way transmission loss is approximately 20 dB (calculated by 0.2 dB/km attenuation). In the prevalent time-division-multiplexing passive-optical-network (TDM-PON) structure, where 64 branches are transmitted over a 20 km distance, the one-way transmission loss can reach 22 dB. Considering the $-14$ dB reflection caused by mismatches between the fiber and the air refractive index at the fault point, the detection sensitivity at the control center should support a path loss of at least $-58$ dB. As the transmission distance or the number of branches increases, the requirement for detection sensitivity will inevitably grow [8,9].

In current methods of fiber fault detection, the fault echo is detected by the photodetector. But the sensitivity of the commercial photodetector cannot support too weak ($< -60$ dBm) laser detection, and the noise generated by the dark current of the photodetector also significantly impacts the quality of signal processing. Although denoising algorithms [10–12], pulse coding detection [13–16], optical frequency-domain reflectometry [17–19], and chaotic light [20,21] can partially enhance the dynamic range of fiber fault detection, they still rely on the photodetector as a fundamental component. While the use of single-photon detectors can push the detection sensitivity to its limit, their inherent limitations and high cost hinder their widespread adoption in the OTDR market [22–26]. Increasing the power of the detection light can accommodate larger path losses in the fiber link without affecting the detector's sensitivity. However, the high-energy pulse induces the nonlinear effects of the fiber, which actually impacts the detection of fault locations.

In our previous research, we introduced a method for improving detection sensitivity and achieving precise fault localization using a modulated laser [27]. Unlike optical frequency-domain reflectometry, this technique is not restricted by the coherent length [28] of the laser and employs the laser itself to receive fault echo, thereby eliminating the sensitivity limitation associated with the photodetector. Theoretical analyzations demonstrated that this method enables the detection of the fault echo in a system with an insertion loss of up to $-118$ dB, which is significantly higher than the detection sensitivity achievable with conventional methods (limited to a maximum of 60 dB of insertion loss with an output power of 0 dBm and a detection sensitivity of $-60$ dBm). In our previous study, we primarily focused on elucidating the reasons for sensitivity enhancement and exploring the influence of various parameters. However, we did not consider the bandwidth and cost limitations of the electronic devices involved in generating the modulation signal during the implementation process. Whether the corresponding requirements can be met and the optimal strategies for parameter settings were not provided.

Therefore, this study extends this method by conducting a comprehensive analysis of the optimal parameter values for modulation signal intensity, frequency sweeping range, and frequency sweeping step. By combining theoretical analysis with experimental investigations, it aims to enhance the detection sensitivity and provide valuable insights into the development of future prototypes. This research offers essential guidance for striking a balance between high-frequency device considerations and measurement performance.

## 2. Principle and Theoretical Model

### 2.1. Principle

The principle of the method for high-sensitivity fault detection using frequency modulation to enhance the external cavity resonant frequency is depicted in Figure 1. The laser receives the echo from the fiber fault point, and due to the presence of optical feedback, the modulation response curve of the laser becomes non-smooth and exhibits periodic oscillations centered around the external cavity resonant frequency ($f$) [29]. By analyzing the periodic fluctuations in this modulation response curve, the position of the feedback can be calculated ($L = c/2nf$, where $c$ represents the speed of light in vacuum and $n$ denotes the refractive index of the fiber). The simplest approach to analyze the periodicity is to perform inverse Fourier transform (IFT) calculations. In this case, the peak corresponding to the fault position in the calculated IFT curve is referred to as the feedback delay signature (FDS). The signal-to-noise ratio (SNR) is defined as the ratio of the FDS height to the noise, while the spatial resolution is represented by the full width at half maximum (FWHM) of the FDS peak.

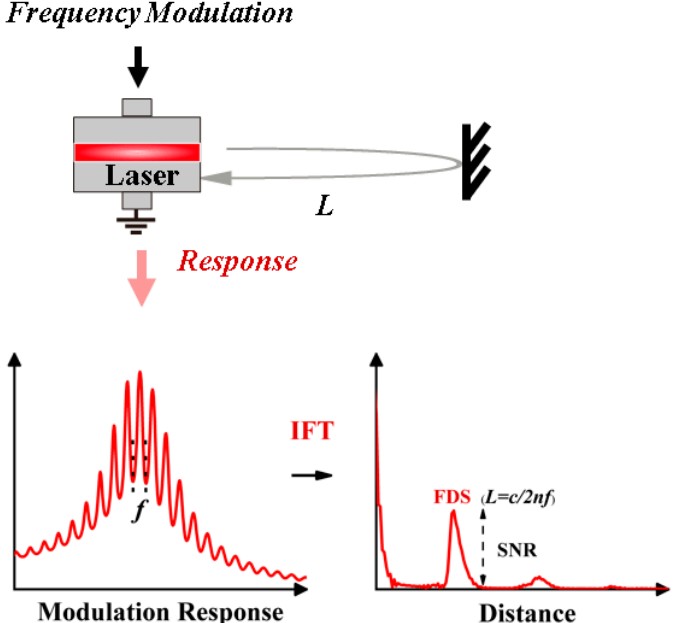

**Figure 1.** Schematic of the modulation-enhanced external cavity resonant frequency in fiber fault detection.

In this method, once the laser is selected, the primary factor influencing the measurement performance is the modulation signal parameters. The frequency sweeping range ($\Delta F$) and the step size ($\Delta f$) of the modulation signal directly determine the sensitivity and spatial resolution of the method. Moreover, these parameters have implications for the measurement time and the selection of electronic devices during the implementation. For instance, a wide range with a small step size in the frequency sweep signal will inevitably increase the detection and subsequent processing time. Excessively high modulation frequency imposes high demand on the high-frequency electronic devices for signal generation and reception. Thus, the practical implementation of this method hinges on the rational selection and configuration of parameters to meet the conventional fiber-fault-detection requirements.

*2.2. Theoretical Model*

Theoretical simulations are conducted based on the Lang–Kobayashi rate equations [30]. By neglecting the spontaneous emission noise of the laser, the rate equations can be expressed as follows:

$$\frac{dA(t)}{dt} = \frac{1}{2}\left\{\frac{\Gamma G_n[N(t) - N_0]}{1 + \varepsilon A^2(t)} - \frac{1}{\tau_p}\right\}A(t) + \frac{k}{\tau_{in}}A(t - \tau)\cos\theta(t) \tag{1}$$

$$\frac{d\phi(t)}{dt} = \frac{1}{2}\alpha\left\{\frac{\Gamma G_n[N(t) - N_0]}{1 + \varepsilon A^2(t)} - \frac{1}{\tau_p}\right\} - \frac{k}{\tau_{in}}\frac{A(t - \tau)}{A(t)}\sin\theta(t) \tag{2}$$

$$\frac{dN(t)}{dt} = \frac{I_m(t)}{qV} - \frac{N(t)}{\tau_n} - G_n[N(t) - N_0]A^2(t) \tag{3}$$

where $A$ and $\varphi$ represent the amplitude and phase of the electric field, respectively, while $N$ represents the carrier density. $\Gamma$ is the optical confinement factor, $G_n$ is the gain coefficient, $\varepsilon$ is the gain saturation coefficient, and $N_0$ is the transparency carrier density. $\tau_n$ and $\tau_p$ represent the lifetimes of carriers and photons, respectively. The round-trip time within the laser cavity is denoted by $\tau_{in}$. The amplitude feedback coefficient is given by $k = (1 - r_0^2)r/r_0$, where $r$ and $r_0$ represent the amplitude reflectivity of the fiber fault and the laser facet, respectively. The feedback phase is given by $\theta(t) = \omega\tau + f(t) - f(t - \tau)$, where $\tau = 2nL/c$ represents the round-trip time between the laser facet and the fiber fault

at a distance *L*. Here, $\omega$, *n*, and *c* denote the angular oscillation frequency, refractive index of the fiber, and speed of light in vacuum, respectively. The linewidth enhancement factor, electron charge, and volume of the laser cavity are denoted by $\alpha$, *q*, and *V*, respectively. The modulation current is given by $I_m(t) = I_b + MD(I_b - I_{th})\cos(2\pi f_m t)$, where $I_b$ and $I_{th}$ represent the bias current and threshold current, respectively, and MD and $f_m$ denote the modulation depth and modulation frequency, respectively. In the fault-detection process, considering that the actual feedback intensity measured is the feedback strength entering the laser after undergoing line losses, the feedback intensity is defined using $k_f = 10\log(r^2)$ (unit: dB). The parameter values are set to be the same as those in reference [27]. As the optical-path loss increases, $k_f$ decreases, and consequently, the FDS height obtained through IFT calculation also decreases. We define $k_f$ corresponding to SNR < 3 dB as the achievable detection sensitivity of this method.

## 3. Numerical Simulation

### 3.1. Sweeping Method

In this method, the practicality implementation is primarily limited by the modulation-frequency sweeping range ($\Delta F$). In the original approach, the modulation response curve exhibits maximum oscillation amplitude near the relaxation oscillation frequency ($f_r$) of the laser. Consequently, $\Delta F$ was defined to extend from $f_r$ towards both sides, as depicted in Figure 2a$_1$. However, the typical $f_r$ of the semiconductor laser is generally within the range of 2.5–5 GHz, making it challenging to directly reduce the frequency of the modulation signal. To address this issue, several feasible approaches can be considered to alleviate the requirements. One option is to utilize a lower $f_r$ laser, or, alternatively, to narrow down the range of $\Delta F$ in order to reduce the demands on the high-frequency devices. Another potential approach involves adopting a different frequency-sweeping implementation by directly sweeping from low to high frequency, as illustrated in Figure 2b$_1$. If this sweeping method can achieve high-sensitivity fault detection within the low-frequency range, it would significantly reduce the requirements for generating and receiving high-frequency signals compared to the conventional $f_r$-centered sweeping approach.

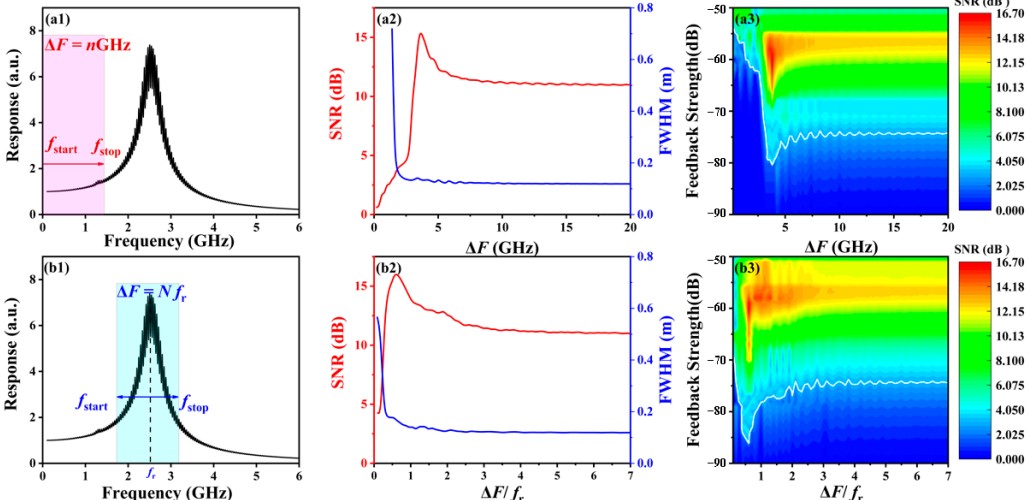

**Figure 2.** Expanding from $f_r$ to both sides as sweeping method: (**a$_1$**) schematic, (**a$_2$**) effects of frequency sweeping range $\Delta F$ on SNR and FWHM, and (**a$_3$**) map of feedback strength and $\Delta F$. Expanding from 100 MHz to high frequency as sweeping method: (**b$_1$**) schematic, (**b$_2$**) effects of sweeping range $\Delta F$ on SNR and FWHM, and (**b$_3$**) map of feedback strength and $\Delta F$.

We compare and evaluate the effects of these two sweeping methods on the measurement results, including the signal-to-noise ratio (SNR) and spatial resolution. In these analyzations, the key parameter settings in the model are as follows: the laser linewidth enhancement factor is set to 5, the cavity length is set to 200 μm, and the facet reflectivity is

set to 0.3. The fault point is set at 23 ns (corresponding to $f$ = 43.5 MHz), under $I_b$ = 1.5 $I_{th}$, $\Delta f$ = 5 MHz, MD = 0.05, and $k_f$ = −60 dB. The resulting modulation response curve is demonstrated in Figure 2a$_1$,b$_1$, with $f_r$ approximately around 2.6 GHz.

Figure 2a$_2$ presents the detection results obtained using the $f_r$-centered frequency sweeping method. In this case, when $\Delta F$ is set to 0.1 $f_r$, the sweeping range extends from 2.47 GHz to 2.73 GHz, covering six oscillation periods and enabling the retrieval of fault information. However, the performance, in terms of the SNR and FWHM, is subpar, measuring 4.26 dB and 0.56 m, respectively. As $\Delta F$ increases, the SNR rapidly improves, while the FWHM decreases sharply until $\Delta F$ = 2$f_r$, following with a gradually stabilizing trend. Throughout this transition, the SNR remains above 3 dB and reaches its peak value of 16.6 dB at $\Delta F$ = 0.6$f_r$. Subsequently, due to the influence of the overall envelope shape of the modulation response curve, the noise in the IFT calculation increases, leading to a decrease in the SNR. Once $\Delta F$ increases to a certain range, a balance is achieved between the oscillation peaks and other noise present in the modulation response curve during the IFT calculation, resulting in a stable SNR of approximately 12.5 dB after $\Delta F$ = 2 $f_r$. Similarly, the FWHM exhibits a similar trend, reaching its minimum value of 0.12 m at $\Delta F$ = 0.6$f_r$ and then remaining stable.

The sensitivity of the detection method in the modulated resonance-enhanced external cavity resonant frequency is directly determined by the SNR magnitude under different parameter settings. To analyze the sensitivity under different $\Delta F$ values, we adjust the $k_f$ and examine the SNR result, as shown in Figure 2a$_3$. While the $\Delta F$ remained constant, the SNR values varied as the feedback strength changed, but the overall trend remained consistent. The maximum SNR was achieved when the feedback strength ranged from −70 dB to −55 dB. The influence of $\Delta F$ followed the same pattern as in the previous study, with a significant impact observed before reaching 2$f_r$. The highest sensitivity of −86 dB was achieved at 0.6$f_r$, followed by a gradual decrease to −75 dB. The region above the white curve in the graph (SNR = 3 dB) corresponds to feedback strengths where FDS peaks can be observed.

We further demonstrate the effects of starting the sweeping from 100 MHz using the same modulation response curve. The SNR and FWHM results are shown in Figure 2b$_2$. Noticeably, the inflection points of both parameter curves shift towards higher values of $\Delta F$. The maximum SNR of 15.5 dB is achieved at $\Delta F$ = 3.6 GHz, followed by a significant decrease to 11 dB and subsequent stabilization. Conversely, the FWHM reaches its minimum within the range of 2.5 GHz and remains relatively stable as $\Delta F$ changes. This phenomenon can be primarily attributed to the low-frequency range before $f_r$, where the laser is less affected by the external cavity resonant frequency, resulting in less pronounced periodic behavior in the modulation response curve. Consequently, the calculated FDS height by IFT is lower, which affects the SNR results. However, the FWHM reaches its optimal value when the SNR is above 3 dB. The relationship between $\Delta F$ and $k_f$ for this sweeping method is illustrated in Figure 2b$_3$, where the sensitivity reaches its lowest value of −80 dB at $\Delta F$ = 4.4 GHz and then stabilizes after 5 GHz. A clear comparison with the results in Figure 2a$_3$ reveals that the sweeping method of the $f_r$-centered mode can achieve the optimal performance at $\Delta F$ = 0.6$f_r$ = 1.56 GHz, allowing for a larger range of detectable feedback strengths. In contrast, the sweeping method starting from lower frequency only achieves a sensitivity of −61 dB at 1.56 GHz.

Considering the high demands on electronic devices for high-frequency wide-range sweeping, it is preferable to have a small sweeping range. Based on the comparative analysis results mentioned above, the sweeping method with $f_r$ as the center expanding towards both sides exhibits several advantages. Firstly, it has a significantly smaller frequency sweeping range and maximum frequency compared to the alternative sweeping method. Secondly, it is less affected by low-frequency components, resulting in higher SNR values. Therefore, in the subsequent research, all frequency sweeping methods are based on the $f_r$-centered sweeping mode.

### 3.2. Optimization of Sweeping Range

The sweeping range centered around $f_r$ faces a challenge due to the impact of bias-current variations on the laser-diode characteristics, including $f_r$ and damping rate. Figure 3a demonstrates the distinct modulation response curves of the laser diode under different bias-current conditions. Increasing the bias current from $1.5I_{th}$ to $10I_{th}$ leads to an increase in $f_r$ from 2.6 GHz to 9.5 GHz. Moreover, the height of the peak at $f_r$ in the modulation response curve decreases, accompanied by a reduction in the amplitude of the periodic oscillations caused by the external cavity resonant frequency. Consequently, if the sweeping range is still determined based on the multiples of $f_r$, the value of $\Delta F$ will increase significantly.

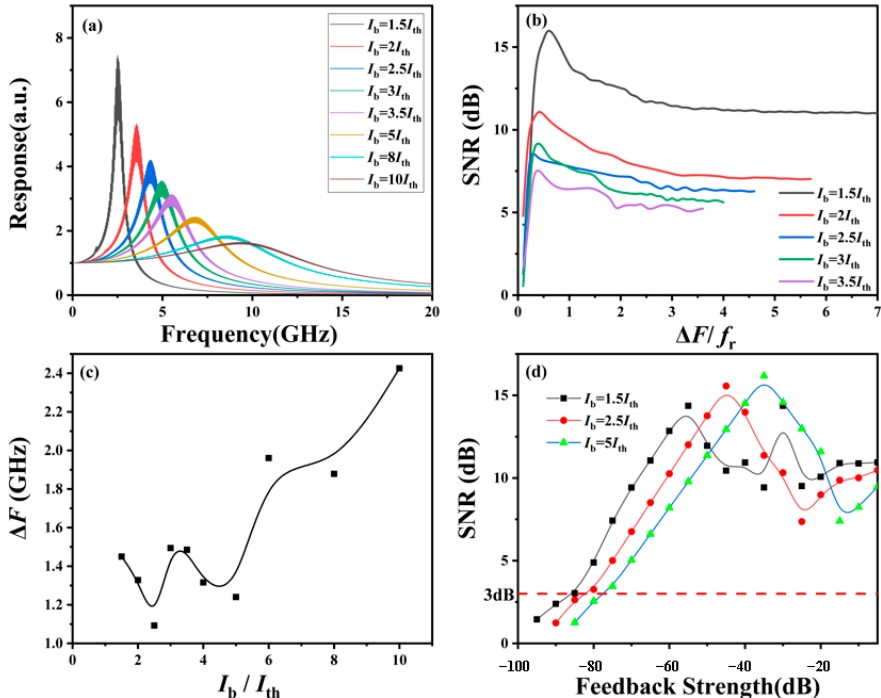

**Figure 3.** (**a**) The laser modulation response curve at different bias currents ($I_b$). (**b**) The effect of frequency sweeping range $\Delta F$ on SNR at different $I_b$. (**c**) The influence of $I_b$ on absolute value of $\Delta F$. (**d**) The influence of feedback strength on SNR at different $I_b$ when $\Delta F = 1.1$ GHz.

To address this issue, we investigated the influence of bias-current variations on the selection of $\Delta F$, building upon the previous section. Figure 3b illustrates the trend of the SNR calculated by IFT under different bias currents as a function of $\Delta F$. The trend follows a similar pattern to previous research results, initially increasing rapidly and then leveling off. The position of the maximum SNR value noticeably shifts towards lower multiples of $f_r$. However, it is important to note that this observation is relative, and a detailed calculation is necessary to determine the absolute change in the sweeping range with increasing $f_r$. Additionally, the change in the curve's height is not consistently in a single direction; it may increase or decrease depending on the specific conditions. Nonetheless, the range of SNR variation consistently remains above 3 dB, satisfying the requirement for fault extraction. Thus, we conducted further analysis to determine the absolute change in the sweeping range.

As shown in Figure 3c, the accurate range of the optimal frequency range $\Delta F$ varies with different $I_b$, exhibiting an overall increasing trend. When the bias current is set to $10I_{th}$, the frequency range reaches a maximum of 2.45 GHz. However, at lower bias currents, there is some fluctuation, and even at a bias current of $2.5I_{th}$, the frequency range achieves the minimum of 1.1 GHz. This indicates that at a bias current of $2.5I_{th}$, the desired effect of fiber fault detection, can be achieved with the minimum scanning range. In Figure 3d,

using this 1.1 GHz frequency range, the variation of SNR with feedback strength was tested at three bias currents. The feedback strength corresponding to an SNR of 3 dB represents the sensitivity of this method. It can be observed that under $\Delta F = 1.1$ GHz, the trend of SNR variation with feedback strength remains consistent, and the difference in sensitivity is less than 2 dB, fully satisfying the detection requirements for fiber faults using this method.

Upon careful observation of the modulation response curves at different $I_b$, it is evident that the number of oscillation cycles included within $\Delta F$ varies for the same multiple of relaxation oscillation frequency. Figure 4 illustrates the modulation response curves for $\Delta F = 0.6 f_r$ at three different bias currents. As the bias current increases, $f_r$ also increases, resulting in a larger number of oscillation cycles being included. However, according to Figure 3b, even though $2.5 I_{th}$ includes more oscillation cycles under the same condition of $0.6 f_r$, its SNR value is significantly lower than that of $1.5 I_{th}$. Based on Figure 4, it can be inferred that in the IFT calculation, the impact of oscillation amplitude on the SNR is much greater than the impact of the number of oscillation cycles. Moreover, the overall envelope of the modulation response curve also varies. This envelope acts as noise in the IFT calculation, affecting the accurate value of the SNR and the sensitivity performance. Therefore, a compromise between the oscillation amplitude and the number of oscillation cycles is necessary. This finding aligns with the conclusion in Figure 3c that the optimal sweeping range is the smallest at $2.5 I_{th}$. Thus, selecting a sweeping range of 1.1 GHz at $2.5 I_{th}$ as the optimal parameter choice is consistent with the objective of achieving good detection performance while reducing the frequency requirements of electronic devices.

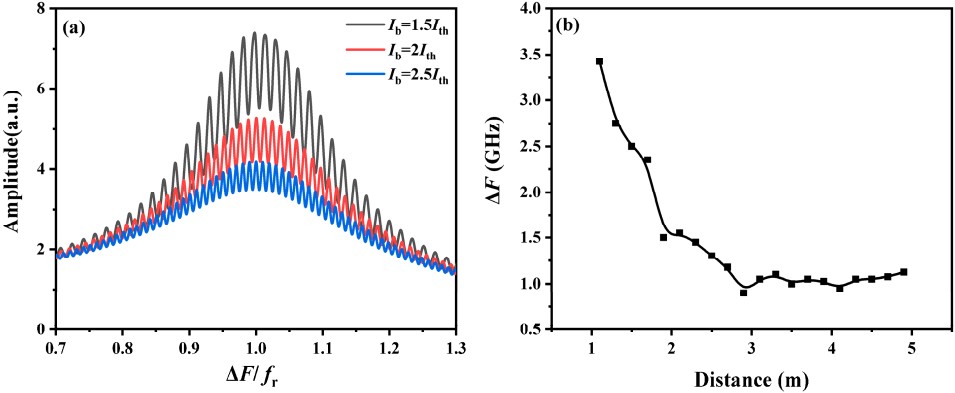

**Figure 4.** (**a**) Modulation response curves for $\Delta F = 0.6 f_r$ at three different $I_b$. (**b**) Influence of different fault distances on the optimal $\Delta F$.

Similarly, the number of oscillation cycles within the sweeping range is determined by the distance to the fault. If the external cavity resonant frequency $f$ corresponding to the fault location is greater than or close to $\Delta F$, the IFT calculation may not accurately determine the frequency associated with the periodic oscillation. For example, when the fault is located at a distance of one meter, assuming a refractive index of 1.5 for the single-mode fiber, the corresponding $f$ is 100 MHz. The chosen sweeping range $\Delta F$ must be greater than this value. However, it requires detailed research to determine the optimal number of oscillation cycles to be included in the sweeping range to achieve the optimal detection performance. Therefore, as shown in Figure 4b, we analyze the optimal $\Delta F$ at different $L$ by the maximum SNR standard (like the process of Figure 2a$_2$). When the external cavity length is one meter, the optimal $\Delta F$ needs to be approximately 3.5 GHz to achieve the maximum SNR. As $L$ increases, the $f$ corresponding to the oscillation cycle in the modulation response curve gradually decreases, leading to a rapid decrease in the required sweeping range. When $L$ reaches three meters, the optimal sweeping range drops below 1 GHz and exhibits minimal changes with further increases in the external cavity length, stabilizing around 1 GHz.

This suggests that once the number of oscillation cycles in the modulation response curve caused by the external cavity resonance reaches a certain threshold, the optimal value of $\Delta F$ will be relatively constant. A balance is achieved between the noise introduced by additional oscillation cycles and the periodic characteristics provided by the IFT calculation. Therefore, in the practical implementation of this method, a three-meter fiber jumper can be added to preset the zero point. This ensures improved SNR performance when $\Delta F$ is small, guaranteeing accurate distance measurement results with $\Delta F = 0.6 f_r$ while maintaining a high SNR.

## 4. Experimental Verification

To validate the optimized parameters obtained from theoretical simulations, we implemented the experimental setup shown in Figure 5. The signal output port of a Rohde & Schwarz ZNB40 vector network analyzer (VNA) was connected to the RF port of a laser diode. The laser output was directly through a polarization controller (PC) and split into two paths by an optical coupler (OC). One path was used as the probing light, which was coupled into the fiber under test (FUT), while the other path was connected to a photodetector (PD) (Finisar XPDV2120RA) to convert the optical signal to electrical signal. The electrical signal feedback was connected to the receiving port of the VNA. This setup allowed us to obtain the modulation response curve of the laser with optical feedback conditions. To precisely control the feedback strength of the laser, we inserted an optical fiber mirror (M) and a variable fiber attenuator (VOA, EXFO FVA-600) in the DUT section. The basic performances of the laser were $I_{th}$ = 7.7 mA, 0.148 mW/mA for the slope efficiency, and about a 1550 nm wavelength. Both the VNA and PD had a bandwidth of 40 GHz. The length of the fiber under test was set to 50 m.

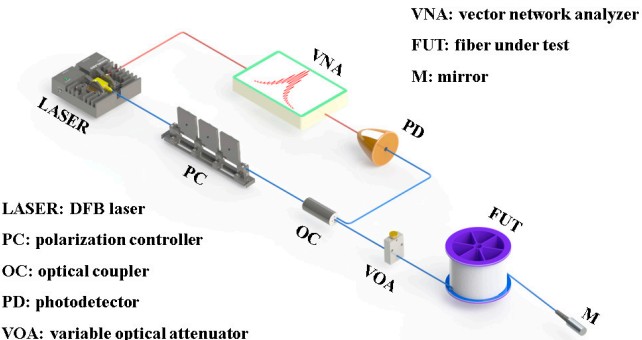

**Figure 5.** The experimental setup of modulation-enhanced external cavity resonant frequency method.

The selected frequency-sweeping method was experimentally verified under $I_b = 1.5\,I_{th}$, where $f_r$ = 4.26 GHz. The variations of the SNR and FWHM with respect to $\Delta F$ in the $f_r$-centered sweeping method are investigated in Figure 6a. The experimental results aligned with the theoretical predictions. The SNR reached its maximum value of 18.3 dB at $\Delta F = 0.6 f_r$ and decreased as $\Delta F$ increased, stabilizing after exceeding $2 f_r$. Similarly, the FWHM exhibited a rapid decrease followed by a plateau, consistent with the theoretical findings. The transition point of the FWHM occurred slightly earlier than that of the SNR, indicating that the FWHM is primarily determined after the IFT calculation, which formed the delayed peak shape. Figure 6b demonstrates the results obtained by sweeping the frequency from low to high. Both the SNR and FWHM showed similar patterns of rapid variations followed by a plateau. The optimal absolute value of $\Delta F$ is about 5.2 GHz. It is noteworthy that the experimental setup exhibited significant background noise, unlike the numerical simulations. Consequently, as $\Delta F$ moved away from $f_r$, it became more susceptible to noise interference. Thus, compared to the simulations, both methods demonstrated earlier inflection points to some extent, and the SNR curve exhibited more fluctuations in the experimental data. It should be pointed out that the value of the SNR in the experiment

is a little higher than the simulation result because of the high-performance VNA drawing the perfect modulation response curve.

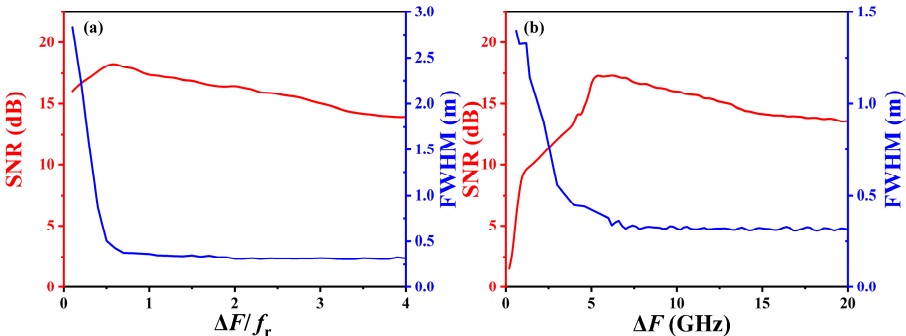

**Figure 6.** (**a**) Experimental verification of effects of frequency sweeping range $\Delta F$ on SNR and FWHM when expanding from $f_r$ to both sides as sweeping range. (**b**) Experimental verification of effects of frequency sweeping range $\Delta F$ on SNR and FWHM when expanding from 100 kHz to high frequency as sweeping range.

In modulation-based frequency sweeping, the choice of step size for the sweeping interval is crucial. Previous studies have highlighted that the step size is equivalent to the precision of the laser's modulation curve and the accuracy of representing the periodic oscillations induced by external cavity resonance. Insufficient sampling points will lead to an inaccurate representation of the periodic oscillation waveform, similar to the concept of the Nyquist theorem in waveform sampling, where an inadequate number of samples cannot fully reconstruct the waveform's characteristics. To address this, the optimal approach is to associate the sweeping step size with the estimated distance to be measured, setting it to one-tenth of the frequency corresponding to that distance. This ensures that each period oscillation in the modulation response curve is represented by ten data points. The results presented in Figure 7a demonstrate that the optimal measurement performance is achieved when there are ten sampling points within each period oscillation caused by the resonance frequency. However, this approach presents a challenge when the distance to be measured increases while the sweeping range is kept fixed. In such cases, the number of sweeping points increases, leading to longer sweeping and subsequent processing times. In such scenarios, a trade-off is necessary, where the sweeping range is shortened based on the desired detection performance.

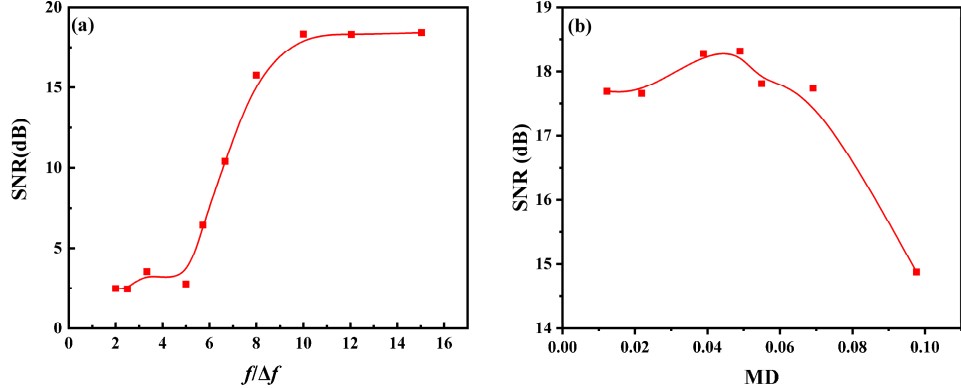

**Figure 7.** Experimental verification of effects of (**a**) frequency sweeping step $\Delta f$ and (**b**) modulation depth MD on SNR.

The current-modulated laser can be classified into two categories: large-signal modulation and small-signal modulation, based on the modulation depth of the modulation signal. Previous research has indicated that the method described in this paper requires

operating the laser in a small-signal modulation state, where modulation depths below 0.1 can achieve high sensitivity for fault detection. However, within the modulation depth range below 0.1, there is still significant room for optimization. Therefore, in this study, we conducted the experimental tests within the modulation depth range of 0.01~0.1, using the optimized parameters obtained from the previous analysis (sweeping range $\Delta F = 0.6f_r$ and bias current 1.5 $I_{th}$). The aim was to measure the SNR results at different modulation depths under the same feedback intensity. The results are shown in Figure 7b. From the graph, it can be observed that at lower modulation depths, the SNR was significantly higher compared to the previous modulation depth of approximately 14.8 dB at 0.1. The highest SNR point occurred at a modulation depth of 0.048, where the SNR reached 18.3 dB. However, increasing the modulation depth beyond this point led to a rapid decrease in SNR.

## 5. Conclusions

This paper presents a practical analysis of the method for high-sensitive fiber fault detection based on modulating resonance-enhanced external cavity resonant frequency. This study explored the optimal frequency-sweeping approach, taking into consideration various practical factors. This research revealed that sweeping from the laser's relaxation oscillation frequency towards both sides requires a smaller sweeping range compared to the traditional sweeping approach from low to high frequency. The detection results achieved a maximum signal-to-noise ratio (SNR) and minimum full width at half maximum (FWHM) when the sweeping range was set at 0.6 times the relaxation oscillation frequency. Furthermore, it was determined that a modulation depth of 0.048 and a sweeping step size of one-tenth of the external cavity resonant frequency were optimal for the method. The choice of bias current significantly affected the relaxation oscillation frequency. Therefore, it is recommended to use a lower bias current for better results. The findings of this study provide crucial insights into the optimal parameter settings for implementing high-sensitivity fiber fault detection. These results serve as a foundation for the subsequent development of practical prototypes.

**Author Contributions:** Conceptualization, T.Z. and A.W.; Methodology, X.L. and Z.S.; Data curation, X.L., M.Z. and H.G.; Writing—original draft, X.L.; Writing—review and editing, T.Z. and Y.G. All authors have read and agreed to the published version of the manuscript.

**Funding:** This research was supported by the National Natural Science Foundation of China (Grant No. 61705160, 62105233), the Natural Science Foundation of Shanxi Province (Grant No. 20210302123183, 20210302124536), the National Defense Basic Scientific Research Project (Grant No. WDYX19614260203), and the development fund in science and technology of Shanxi Province (YDZJSX2022A010).

**Institutional Review Board Statement:** Not applicable.

**Informed Consent Statement:** Not applicable.

**Data Availability Statement:** The data presented in this study are available on request from the corresponding author.

**Conflicts of Interest:** The authors declare no conflict of interest.

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
