# Peer review of "Parameter Optimization for Modulation-Enhanced External Cavity Resonant Frequency in Fiber Fault Detection"

_photonics, doi:10.3390/photonics10070822_

Round 1

Reviewer 1 Report

In this work, based on the previously proposed modulation-enhanced external cavity feedback resonance method, the authors optimized the parameters to acquire good performance in high sensitivity fiber fault detection. Including the sweeping method, sweeping range, sweeping step, and modulation depth. These optimizations provide a guidance for practical application of this method. The manuscript is innovative and important, while I suggest the authors address the following issues:

(1)       In Figure 3(d), why the SNR decreases after the peak point, and then demonstrates a fluctuation trend? Why the SNR cannot keep stable at the maximum value.

(2)       In the simulation results, the minimum absolute value of frequency sweeping range is 1.1 GHz. However, in the experimental setup, the minimum frequency scanning range is about 2.5GHz (0.6fr = 2.55 GHz), and higher Ib will continue increase the range. Please explain the reason of the different values.

(3)       The author realizes the frequency sweep by applying different currents to the DFB laser source. In addition to the frequency sweeping range, has the author considered the influence of the frequency sweeping linearity on the system performance? The sweeping may change the laser phase [e.g. Nature Comm. 12 (1), 6716 (2021); Phys. Rev. Lett. Physical Review Letters 130(15), 153802 (2023)], can the authors briefly discuss this point?

(4)       Usually, for signal testing on optical fiber links, typical performance parameters should include response sensitivity, spatial resolution, sensing distance, etc [ e.g. Research 2021, 5612850]. In this article, the author uses SNR and FWHM to express system performance, which I think is not intuitive and easy to understand.

(5)       Line 154, “the laser linewidth is set to 5”, please give a unit.

(6)       There are some mistakes or inappropriate expression need to be corrected. For example:

1)          The symbol of frequency in Figure 1 is inconsistent, f or fr? It should be confirmed.

2)         In Figure 4, the relaxation oscillation frequency is not indicated as the center.

3)         The format of references should be unified. In reference [10] the journal is “J. Light. Technol.”, but in [26] is “J. Lightwave Technol.”;

4)         In Line 51-52, the space should be use correct before Punctuation or after text.

In a nutshell, this is a pretty interesting investigation, I recommend to accept this manuscript after a minor revision.

Generally good.

Reviewer 2 Report

This manuscript revealed that optimal detection performance of the modulation-enhanced external cavity resonant frequency method that utilizes a laser for fault echo reception. The results are well presented. Below are several concerns:

1) The authors wrote: However, at lower bias cur- 240 rents, there is some fluctuation, and even at a bias current of 2.5Ith, the frequency range 241 achieves the minimum of 1.1 GHz. What is the reason of the fluctuation.

2) The length of the fiber under test was set 306 to 50 m. Why 50 m was chosen

N/A
